# Overview of the MEMS Pirani Sensors

**DOI:** 10.3390/mi13060945

**Published:** 2022-06-14

**Authors:** Shaohang Xu, Na Zhou, Meng Shi, Chenchen Zhang, Dapeng Chen, Haiyang Mao

**Affiliations:** 1Institute of Microelectronics of Chinese Academy of Sciences, Beijing 100029, China; xushaohang@ime.ac.cn (S.X.); zhouna@ime.ac.cn (N.Z.); shimeng@ime.ac.cn (M.S.); zhangchenchen@ime.ac.cn (C.Z.); chendapeng@ime.ac.cn (D.C.); 2University of Chinese Academy of Sciences, Beijing 100049, China; 3Jiangsu Hinovaic Technologies Co., Ltd., Wuxi 214135, China

**Keywords:** Pirani sensors, MEMS, vacuum, thermal conductivity, functional materials

## Abstract

Vacuum equipment has a wide range of applications, and vacuum monitoring in such equipment is necessary in order to meet practical applications. Pirani sensors work by using the effect of air density on the heat conduction of the gas to cause temperature changes in sensitive structures, thus detecting the pressure in the surrounding environment and thus vacuum monitoring. In past decades, MEMS Pirani sensors have received considerable attention and practical applications because of their advances in simple structures, long service life, wide measurement range and high sensitivity. This review systematically summarizes and compares different types of MEMS Pirani sensors. The configuration, material, mechanism, and performance of different types of MEMS Pirani sensors are discussed, including the ones based on thermistors, thermocouples, diodes and surface acoustic wave. Further, the development status of novel Pirani sensors based on functional materials such as nanoporous materials, carbon nanotubes and graphene are investigated, and the possible future development directions for MEMS Pirani sensors are discussed. This review is with the purpose to focus on a generalized knowledge of MEMS Pirani sensors, thus inspiring the investigations on their practical applications.

## 1. Introduction

Vacuum science has wide applications in many fields [1,2,3,4,5,6,7], such as semiconductor industry [8,9], aerospace [10,11] and military [12,13]. In semiconductor industry, the growth of pure silicon requires a vacuum environment to reduce molecular contamination, besides, micromachining steps including reactive ion etching (RIE), chemical vapor deposition (CVD) and physical vapor deposition (PVD) also need to be completed under vacuum with a clean environment and reduced molecular interference. To monitor the vacuum information in these environments, vacuum sensors are demanded. Additionally, many devices require vacuum encapsulation to optimize performance and reliability [14], such as MEMS gyroscope sensors [15], high-end micro-accelerometers [16], uncooled infrared focal plane arrays (UFPA) [17,18,19], etc. These devices need to operate in a vacuum environment either to reduce gas damping of their moving parts or to decrease gas thermal conduction, thus improving their performance. It is obvious that the research on vacuum sensors is of great importance.

In 1906, Marcello Pirani discovered that the heating current needed for melting metals in a vacuum environment was reduced with a decrease in air pressure. In other words, when air density in a chamber is decreasing, the heat conduction through the gas declines simultaneously [20]. This phenomenon was named the Pirani effect. Correspondingly, devices based on such an effect through the change of heat conduction to detection of gas pressure is named Pirani sensors [21].

After a long period of development and improvement, Pirani sensors are now widely used in many fields involving vacuum monitoring [22,23,24,25,26,27,28], such as automotive industry, food processing and others. Although traditional Pirani sensors are very classic, they are large, costly, laborious and have difficulty in mass production [29]. In view of this, miniaturization of Pirani sensors is of significance. In recent decades, the development of MEMS technology has greatly promoted the miniaturization of traditional sensors [30,31,32,33,34,35,36,37,38,39,40,41,42,43]. The miniaturized Pirani sensors, also known as the MEMS Pirani sensors, can not only reduce the weight and size of the devices, but also can cut down power consumption and production costs [44,45,46,47,48,49,50,51,52,53,54,55]. Moreover, the MEMS technology can facilitate integration of Pirani sensors with other devices on one chip [56,57,58]. With these features, the MEMS Pirani sensors have been adopted in areas where traditional Pirani sensors cannot reach, including the micro vacuum chamber systems and the vacuum packaged chips [59]. MEMS Pirani sensors can be divided into various types according to the measurement principles, each with different properties and advantages. However, they have not been systematically discussed and compared so far.

In this review, the MEMS Pirani sensors are systematically introduced, as shown in Figure 1. The mechanism, configuration, material, and performance of different types of MEMS Pirani sensors are compared and summarized, including the ones based on thermistors, thermocouples, diodes and surface acoustic wave (SAW). Besides, the development status of some novel Pirani sensors taking uses of functional materials such as nanoporous materials, carbon nanotube (CNT) and graphene is also introduced. By summarizing the different type MEMS Pirani sensors, it is found that, although the MEMS technology has greatly contributed to the development of Pirani sensors, there are still serious problems that limit the practical applications of these sensors, such as detection range not being wide enough and that the fabrication process of some devices is not compatible with the CMOS process, which makes the production of MEMS Pirani sensors more costly and relatively low in performance. Therefore, it is urgent to develop new structures and find new materials for MEMS Pirani sensors that can be mass produced based on simple preparation processes while at the same time extending the detection range, enabling the sensors to be useful in practical applications. This review provides a perspective on the possible future directions and applications of MEMS Pirani sensors.

## 2. Thermistor-Based Pirani Sensors

For a thermistor-based MEMS Pirani vacuum sensor, the heater is also a thermistor, the resistance of which is a function of its temperature, while the change in temperature is closely related to the air pressure [60,61,62]. When air pressure varies, heat loss through the air is changed, which, as a result, leads to temperature variation of the thermistor, accordingly, resistance of the thermistor is also changed. By measuring the voltage at both ends of the heater, with the applied current flowing through, resistance of the heater can be obtained. Meanwhile, based on the temperature coefficient of resistance (TCR) of the thermistor, its temperature and the air pressure can be obtained correspondingly [63,64,65]. The higher the TCR, the more sensitive the thermistor (also the heater) to the change of temperature, and thus the higher sensitivity of the Pirani sensor can be achieved. According to the direction of heat conduction, there are two types of thermistor-based MEMS Pirani sensors, including the vertical heat transfer configuration and the lateral heat transfer configuration.

### 2.1. Vertical Heat Transfer Configuration

In Pirani sensors with a vertical configuration for heat transfer, the heater and heat sink are vertically distributed [66,67,68,69,70,71]. In these devices, the heater is usually made of NiCr [66], tungsten [67], platinum [70], polysilicon [60], etc. Generally, there are two design concepts for the vertical heat transfer configuration, which are microbridges and dielectric membranes. The heater with a microbridge structure is usually distributed in the device as a suspended strip or a wire resistor, while the heater with a dielectric membrane structure is usually distributed as a serpentine resistor patterned on a dielectric membrane. On one hand, the dielectric membrane is used as a supporting structure, and the area of the heater can be made relatively large. On the other hand, the dielectric membrane can also reduce the solid heat transfer between the heater and heat sink, this is because the dielectric membrane is located between the heater and heat sink, which can avoid direct contact of the two structures. Since in a microbridge-based device there is no membrane supporter for its heater, the gap between the heater and the heat sink is quite small, meanwhile, in order to maintain structural stability, the area of the heater is relatively small. In short, performance of the microbridge-based device is usually limited by the small size of the heater, but the gap between the heater and the heat sink can be relatively close. The heater located on a dielectric membrane is relatively large in size; however, the gap between the heater and the heat sink cannot be controlled when very small. The detailed effect of gap and size of the heater on device performance will be discussed afterwards.

The devices based on microbridge structures are more suitable for high pressure detection [60]. As the gap between the heater and the heat sink in a microbridge-based device is very small, with the circumstance of high pressure, there are a large number of gas molecules, therefore, the molecules can only have a small mean free path [26]. As a result, it is likely for the gas molecules to carry heat from the heater to the heat sink, thus it is conducive to the conduction of heat through the gas molecules, and therefore, such a device with a small gap is beneficial for detecting high pressure conditions [66]. However, in a dielectric membrane structure, as there is a layer of dielectric membrane, the gap in between cannot be as small as that in microbridge devices. On this basis, the dielectric membrane-based devices are not suitable to detect the high-pressure conditions. Conversely, the sensitive areas of the dielectric membrane devices can be designed much larger when compared with the microbridge devices, which makes these devices more suitable for low pressure detections [60]. This is because under low pressure conditions, there are few gas molecules, thus the mean free path of gas molecules is much larger, thus the size of the gap has little effect on movement of gas molecules from the heater to the heat sink, however, in such a case, the size of the heater will become a main influencing factor [22]. The larger the heater is, the more gas molecules can come into contact with the heater and take heat away, thus it is more conducive to the conduction of heat through the gas [72]. However, for a suspended microbridge structure with a large size, it is more likely to collapse. As a dielectric membrane can also function as a supporting structure, the heater size can be prepared much larger, in view of this, the dielectric membrane-based devices are suitable for low pressure detection.

Figure 2a shows the schematic structure of a thermistor-based Pirani sensor with a vertical heat transfer configuration. As demonstrated in the diagram, the heater and the heat sink are longitudinally distributed, and the heater is located over the heat sink. As discussed previously, the microbridge device with a small gap exhibits excellent performance in high pressure detections. In order to achieve sufficient sensitivity in a high-pressure range, the gap between the heater and the heat sink should be controlled at the level of nanometers. Additionally, for the dielectric membrane-based devices, the interaction area between the heater and the gas molecules is larger, it will greatly improve the heat conduction of the gas, therefore, dielectric membrane devices can reach an outstanding performance in low pressure detections. Another reason for the dielectric membrane-based devices to improve performance in the low-pressure detection is that they are able to reduce the solid heat conduction. As in a low pressure, only few gas molecules are involved in the heat conduction, if there is no dielectric membrane blocking the direct contact between the heater and the heat sink, almost all the heat will quickly dissipate through the solid heat conduction (through the heat sink), thus the relationship between the output of the device and the pressure cannot be detected. With the dielectric membrane between the heater and the heat sink, the solid heat conduction can be effectively blocked, thus heat conduction through gas cannot be ignored but be used for pressure detection. In short, even if the air pressure change is small at low pressure condition, the device can still detect the change of the air pressure.

Khosraviani et al. reported a Pirani vacuum sensor based on a microbridge structure, in which NiCr was adopted to construct the heater as shown in Figure 2b [66]. To prepare this device, firstly, an amorphous silicon layer grown by Plasma Enhanced Chemical Vapor Deposition (PECVD) was used as a sacrificial layer. Subsequently, gas phase etching involving xenon difluoride (XeF_2_) was used to release the sacrificial layer, thus the suspended microbridge with a nanoscale gap was formed. The testing results are demonstrated in Figure 2c. As is shown, the narrow gap between the heater and the heat sink could extend the testing limit in a high-pressure range and improve sensitivity of the device. The device with a gap of 50 nm achieved a response range from 10^2^ to 7 × 10^5^ Pa, which was wider than that of the device with a 100 nm gap, whose response range was from 10^2^ to 10^5^ Pa. Besides, the former also had higher sensitivity than the latter.

Zhang and coworkers presented a Pirani sensor based on a dielectric membrane structure with four supporting beams, as is shown in Figure 2d [73], the heater in this device was made of polysilicon. The testing results are shown in Figure 2e. As is illustrated, the sensitive range of the device was 10 to 10^5^ Pa. In order to reduce the solid heat conduction and to broaden the sensitive range of the device, Zhang et al. reported another dielectric membrane-based sensor with six supporting beams. Similarly, polysilicon was used for constructing the heater, as shown in Figure 2f [74]. On basis of the four supporting beams used in their previous work, the authors adopted two other narrow and short beams with low thermal conduction in the structure, which were able to further improve the mechanical strength of the device, so that the other four beams could be longer to efficiently reduce the total solid heat conduction, and consequently, to achieve a lower limit of the detectable pressure range. The output voltage of the device versus air pressure under the condition of different heating current is shown in Figure 2g, the results illustrate that the sensitive range for this device was 10^−1^ to 10^5^ Pa.

To demonstrate the effect on sensitivity and sensitive range of the heater area in the dielectric membrane device, Lefeuvre et al. prepared three Pirani sensors based on dielectric membranes with different heater areas, as shown in Figure 2h [72]. The heaters of the three devices were all made of aluminum, likewise, heaters in these devices were all heated with a constant current, variations of the resistance versus the air pressure are shown in Figure 2i, which illustrate that the device with a larger heater area has a larger sensitivity and a wider sensitive range than that with a smaller heater area.

J.J. van Baar et al. have developed a thermistor type Pirani sensor with a vertical heat transfer configuration, a gap between the heater and the heat sink in a V-shaped groove as shown in Figure 3 [75]. This Pirani sensor could detect pressure by measuring the temperature distribution of the heater rather than the average temperature change of the heater. The advantage of this Pirani sensor compared to the previous vertically structured thermistor type sensors is that it does not need to take into account the TCR magnitude of the heater materials, thus implicitly taking into account the heat loss of the substrate [76,77].

### 2.2. Lateral Heat Transfer Configuration

Pirani sensors with a lateral heat transfer configuration are generally realized requiring fewer steps of photolithography and no corrosion processes when compared with the vertical heat transfer configuration. In such a device, the heater and the heat sink are usually prepared from the same material layer. Therefore, this type of Pirani sensor has advantages including ease to manufacture [78,79,80,81].

Figure 4a shows the schematic structure of a thermistor-based Pirani sensor adopting the lateral heat transfer configuration. As shown in the scheme, the heater and the heat sink are horizontally distributed, and the heat sink is distributed on both sides of the heater. Gas heat conduction is transferred horizontally from the heater to the heat sink. Figure 4b illustrates a sensor of this type reported by Jiang et al. [78], in this sensor, the heater and the heat sink were both prepared from a single crystal silicon layer. The heater was designed in a serpentine shape and fixed between two comb-shaped heat sinks. The relationship between the thermal impedance of the device and the pressure is shown in Figure 4c, the results illustrate that the sensitive range of the device was from 1 to 1000 Pa. In addition, Topalli et al. fabricated two Pirani sensors with the lateral heat transfer configuration by using a dissolved-wafer process (DWP) and a silicon-on-glass (SOG) process, respectively, as shown in Figure 4d,e [79]. The heaters and the heat sinks of the two devices were all made of polysilicon. Importantly, Topalli et al. performed a detailed study on the two fabrication techniques, they also tested and compared in detail the performance of the devices the two techniques achieved, the resistance change versus air pressure of the two devices is shown in Figure 4f. The results show that the sensitive range of the DWP device was from 10 to 2000 mTorr (about 1.333 to 266.6 Pa, 1 Torr = 133.3 Pa). While the sensitive range of the SOG device was from 50 to 5000 mTorr (about 6.665 to 666.5 Pa).

## 3. Thermocouple-Based Pirani Sensors

A thermocouple-based Pirani sensor works based on the Seebeck effect [82], when the hot ends are heated, at the cold ends there forms a voltage, as is shown in Figure 5a. This type of Pirani sensors is usually composed of several pairs of thermocouple strips [83,84,85,86,87,88]. The thermocouple strips are connected in series to form a thermopile, where the hot ends are heated by a heater, while the cold ends are located on heat sinks, which keeps the temperature of the cold ends consistent with the ambient temperature. During operation, the heater is applied with a constant heating power, which leads to a temperature rise at the hot ends of the thermopile, and then the temperature variation is converted into an electrical signal according to the Seebeck effect. When pressure varies, the gas heat conduction is changed accordingly. Since the heat conduction of gas is a part of the total heat conduction, therefore, if the gas heat conduction changes, the proportion of solid heat conduction from the hot ends to the cold ends changes correspondingly, which in turn affects the temperature change between the hot ends and cold ends. Based on this principle, the relationship between the pressure and the output electrical signal is detected. Figure 5b shows a schematic structure of a thermopile, usually, a thermocouple-based Pirani sensor is designed and manufactured on basis of this structure.

Sun et al. explored a thermocouple-based Pirani sensor that was fully compatible with the complementary metal oxide semiconductor (CMOS) process, as shown in Figure 5c [89]. The device consists of a polysilicon heater and 38 N-polysilicon/Al thermocouples, the thermocouples were suspended on a cavity, and the cavity was realized by using XeF_2_ front-side dry-etching process. The heater was prepared from poly-Si. During operation of the device, the heater was applied with a constant power. The relationship between the output voltage of the device and the pressure is shown in Figure 5d, as is illustrated, the sensitive range of the device was 5 × 10^−3^ to 10^5^ Pa.

Lei et al. designed a Pirani sensor based on n-poly-Si/p-poly-Si thermocouples, which were located over a cavity prepared by a front etching process involving XeF_2_, as is shown in Figure 5e [90]. In the device, there was no heater, therefore, external apparatus was required for heating up the hot ends. During operation, the external heat radiation source provides a constant heating power to the heat absorption area of the device. With the Seebeck effect, the output voltage of the device versus the gas pressure was measured, as shown in Figure 5f. As is illustrated in this figure, the sensitive range of the device was 10^−1^ to 10^4^ Pa. Besides, sensitivity of the device increases with the rising of heat radiation power from the external apparatus.

## 4. Diode-Based Pirani Sensors

To achieve pressure measurement capability of the diode-based Pirani sensors, the temperature coefficient of diodes is taken into use. The diode-based Pirani sensors usually consist of p/n-junctions while requiring no special heat-sensitive materials, as shown in Figure 6a. When pressure changes, voltage can be generated at the two ends of the diode, thus pressure detection is achieved. For this type of Pirani sensors, their manufacturing process is quite simple and is CMOS compatible, which allows easy integration of the sensors with other devices/circuits. On this basis, the diode-based Pirani sensors have attracted much attention [91,92,93].

Figure 6b is a three-dimensional diagram of a diode-based Pirani sensor proposed by Wei et al. [91]. In this device, a series of diodes were connected and manufactured on a silicon on insulator (SOI) wafer. The device was composed of a sensitive area, a cantilever, a frame and the Si substrate. The diodes in series were embedded in the sensitive area, and were used for heater and pressure detection simultaneously. When a constant current was applied to the diode, the temperature of the diode increases due to the self-heating effect. Subsequently, heat will be conducted from the diode to the environment, which in return will cause temperature decrease in the diode until an equilibrium temperature was reached. As previously mentioned, the heat transfer was dependent on environmental pressure. Therefore, when the pressure changes, the equilibrium temperature of the diode will also be changed, and based on the temperature coefficient of the diode, the change in output voltage will be obtained. By connecting the diodes in series, the temperature coefficient of the diodes was further increased, so that the device achieves a higher performance. Figure 6c shows the measured relationship between the output voltage of the diodes and the pressure. The result show that the sensitive range for the sensor with six diodes in series can reach 10^−1^ to 10^4^ Pa.

Figure 6d is a schematic diagram of a diode-based Pirani sensor reported by Kimura et al., and Figure 6e is a micrograph of the device [92]. The device was composed of a heater and two p/n-junction diodes, labeled as Th-A and Th-B. The working principle is similar to that of a thermistor-based sensor on a micro-bridge. Th-A was used to measure temperature Ta of the heater and to control its temperature as well. Meanwhile, Th-B was used to measure the temperature Tb, which was sensitive to pressure. The ambient temperature Tc of the surrounding gas was measured by a third thermistor-like diode Th-C located at the edge of the chip. The temperature Ta of the heater and Th-A was kept constant by feedback control through Th-C. Then, output voltage difference V_AB_ (V_AB_ = V_Th-B_−V_Th-A_) between Th-B and Th-A under different pressure conditions was measured. Figure 6f shows the relationship between pressure and output voltage at a constant ambient temperature of 25 °C. The output voltage refers to the output voltage difference between the diode-thermistor Th-A and Th-B. In this experiment, the temperature of the heater Ta was maintained at 85 °C. When the vacuum was at a high level, the temperature of Th-B will be close to the value of Th-A, because the heat dissipation from Th-B to the ambient gas was reduced. On the other hand, at a vacuum degree close to the atmospheric pressure, the temperature of Th-B formed in area B decreases, thus the temperature difference between Th-A and Th-B increases, as the heat from area B was dissipated into the surrounding environment. This thermal vacuum sensor with the novel structure has been proved to be able to measure a very wide sensitive range from 2 × 10^−3^ to 10^5^ Pa.

## 5. SAW-Based Pirani Sensors

For traditional and the thermistor-based Pirani sensors, they work because their resistance changes with the varying pressure, while for a SAW-based Pirani sensor, it is the wave frequency that changes with the varying pressure [94,95,96]. As is known, propagation frequency of SAW is very sensitive to parameters such as temperature, pressure and humidity of the surrounding environment. For instance, when pressure of the environment is changing, the number of gas molecules in the environment varies correspondingly, which as a result, will affect the heat dissipation from the heater to the environment, consequently, the temperature variation trend of the heater is changed. Further, the temperature variation will modify the elastic coefficient and density of the piezoelectric crystal material, on surface of which SAW usually propagates, thus frequency of the waves will be affected.

Compared with other piezoelectric materials, Lithium Niobate (LiNbO_3_) has a much higher temperature frequency coefficient, and is widely used in SAW-based Pirani sensors [97]. Such a LiNbO_3_-based Pirani sensor can detect small pressure variation even down to 10^−3^ Pa [98].

Figure 7a show the schematic structure of a SAW Pirani sensor, which uses a layer of piezoelectric material as its substrate, and on its back side, there is a metal layer. In the device, the transmitting interdigital transducer (IDT) and receiving IDT are prepared on the piezoelectric substrate using e-beam lithography. The IDTs of both types are composed of two interlocking comb-shaped metal coatings. Herein, the transmitting IDT is used to convert the electrical pulses (generated by the radio frequency circuit) into SAW, and then the SAW propagates to the receiving IDT, which subsequently converts the SAW signals into electrical signals. In the measurement, the temperature of the piezoelectric material changes correspondingly with the variation of pressure, which leads to change of SAW propagation frequency, as well as the output electrical signals from the receiving IDT. Besides, the heater in such a device is usually located around the IDTs for achieving uniform heating of the device. Figure 7b demonstrates the structure of the IDTs.

Figure 7c is a typical structure of a SAW-based Pirani sensor designed by Singh et al. [98]. In this work, the authors stated that, the lower the emissivity of the substrate, the higher sensitivity the device can achieve. By depositing an Al layer at backside of the piezoelectric material, sensitivity of the device can be significantly improved due to the low emissivity of Al. Frequency of SAW signals versus the pressure of the two devices with and without the Al layer is illustrated in Figure 7d. The results demonstrate that the Al layer can help with the enhancement of sensitivity, besides, it can also help to enlarge the sensitive range, based on this approach, the low limit of the detection range can reach 10^−3^ Pa.

## 6. Functional Material-Based Pirani Sensors

In general, the sensing elements used in Pirani sensors are made of semiconductor, metal or piezoelectric materials. In recent years, Pirani sensors involving functional materials have been frequently reported, including the sensors adopting nanoporous anodic aluminum oxide (AAO) [99], carbon nanotubes (CNTs) [100,101,102] and graphene [103,104,105]. With these functional materials, the Pirani sensors are able to extend the sensitive range. Furthermore, this novel approach also helps to broaden applications of the Pirani sensors.

Jeon et al. reported a Pirani vacuum sensor based on nanoporous AAO [99]. The device consists of a silicon substrate, an AAO membrane and a nickel heater. The AAO membrane was suspended on the silicon substrate as a supporter, on which a heater was distributed in a serpentine shape. As there was a gap between the AAO membrane and the substrate, which functions as a heat sink, thus heat conduction was promoted. The schematic of this device is shown in Figure 8a, as is demonstrated, there were vertical through-pores distributed all over the membrane, and the porosity can be adjusted between 12–30%. Ascribed to the nanopores, the effective sensing area was increased, it enhances the gas heat conduction, and the solid heat conduction was reduced due to the presence of a membrane, so that a lower detection limit of pressure is achieved. For such a device, its resistance variation with air pressure is shown in Figure 8b. The results show that as the porosity increases, the device can detect a lower level of pressure, and a porosity of 25% for this device can detect a pressure as low as 10^−1^ mTorr (about 10^−2^ Pa). This was the first time a Pirani sensor adopted AAO as its sensing element and the supporting membrane, in this work, porosity of the nanopores could be controlled and adjusted to achieve optimized performance.

CNTs have been widely used in various electronic devices due to their unique thermal, electrical, chemical and mechanical properties [106,107,108]. This type of material provides huge opportunities for the miniaturization and intelligentization of electronic devices. Ascribed to their nanoscale size, high thermal and electronic conductivity, CNTs are quite suitable as a suspending heater for use in Pirani sensors. The contact area between the CNT heater and the heat sink is very small since the diameter of the CNTs is in nanoscale, resulting in a significant reduction of solid heat conduction, therefore the lower-pressure range can also be detected. In addition, CNTs usually have a large TCR and a high thermal conductivity, which can lead to large sensitivity and rapid thermal response of the device. Moreover, the excellent electrical conductivity of the CNTs can pull down power consumption of Pirani sensors to a low level. Kaul et al. designed a Pirani sensor based on single-walled nanotube (SWNT). The structure of this device is shown in Figure 8c [100], as is shown, the SWNT was suspended between two Au/Cr electrodes, herein, the SWNT was prepared by CVD growth. Figure 8d shows the results of the output characteristics of the device, the current of the device changes as a function of air pressure under different bias voltages. The results show that the sensitive range of the device reaches 10^−6^ to 760 Torr (about 10^−4^ to 10^5^ Pa).

The emergence of graphene has initiated a new field of electronic devices based on two-dimensional materials [109,110,111,112,113]. Due to its excellent properties, such as large surface area, high electrical conductivity, and good chemical stability, graphene has gradually been applied to various types of sensors, including the Pirani sensors. As graphene is a thin two-dimensional material, so the solid contact area between the two-dimensional graphene and the heat sink is very small, thus the solid heat conduction between the graphene heater and the heat sink is reduced, therefore, compared with the solid heat conduction, the gas heat conduction cannot be ignored even under low pressure, in view of this, the sensitivity range of this sensor is extended. Figure 8e is a schematic diagram of a Pirani sensor designed by Romijn et al. [103], the device uses a multi-layered graphene strip suspended on a heat sink as a heater. As is reported, the material was grown using a selective and transfer-free CVD method. Figure 8f illustrates the resistance variation of the device with pressure in different gas environments. Such a device can also be used as a gas sensor to identify gas components, on that account, the application range of the graphene-based Pirani sensors can be expanded.

Figure 8g shows a Pirani sensor based on a reduced graphene oxide nanocomposite with hollow αFe_2_O_3_ (rGO/α-Fe_2_O_3_), developed by Shirhatti et al. [104]. Herein, rGO/α-Fe_2_O_3_ with a high TCR was used as the sensing element. With such a device, when the vacuum degree increases due to the lack of gas molecules functioning as a heat transfer medium, the heat transferring to the surrounding environment was reduced, leading to a temperature rise of the rGO/α-Fe_2_O_3_, which, as a result, changes the charge concentration in the material, thereby affecting conductivity of the heater as well as resistance of the device. On the other hand, the multi-shell spherical structure of the rGO/α-Fe_2_O_3_ increases its surface area which affects the conductivity of the rGO/α-Fe_2_O_3_. This is because, the large surface of the rGO/α-Fe_2_O_3_ was conducive to absorbing gas, and the absorbed gas was also conducive to the adsorption of electrons for the rGO/α-Fe_2_O_3_, therefore, it affects the conductivity of rGO/α-Fe_2_O_3_. Figure 8h shows the resistance variation rate of the rGO/α-Fe_2_O_3_ with the air pressure. The result shows that in a pressure range from 4 × 10^−6^ to 10^3^ mbar (4 × 10^−4^ to 10^5^ Pa), the device exhibits an excellent sensitivity.

By summarizing the various type MEMS Pirani sensors, we have collated their properties in Table 1. As shown in Table 1, for the thermistor-type MEMS Pirani sensors, these with vertical thermistors have a wider pressure measurement range, a smaller active area and lower power consumption than those with lateral thermistors; however, the latter type has a higher sensitivity and a simpler preparation process than the former ones. It should be noted that the fabrication processes of Pirani sensors with both vertical thermistors and lateral thermistors are compatible with the CMOS process. Similarly, the process of the thermocouple type MEMS Pirani sensors is also compatible with the CMOS process. Moreover, this type of device has a lower detection limit smaller than those of the thermistor type, and can detect pressure at lower ranges, but it has a more complex structure and thus are more difficult to prepare. The diode type sensors have a similar lower detection limit with that of the thermocouple ones, and can also achieve a wider range of pressure detection, besides, they have lower power consumption, but their sensitivity is low, in addition, they have relatively larger active areas, and moreover, the process complexity of these sensors is increased as preparation of p-n junctions is required. When compared with those based on thermocouples and diodes, the SAW type sensors can achieve a smaller lower detection limit and have a wider pressure detection range, but their active areas are even larger than diode type Pirani sensors, and power consumption is also higher, besides specific piezoelectric materials are required, moreover, their preparation processes are more difficult and are usually not compatible with CMOS process, which is not conducive to integration. For the novel Pirani sensors integrated with functional materials, they have a much smaller lower limit of detection for pressure, a wider detection range, a smaller active area and lower power consumption due to the introduction of nanomaterials, and a relatively high sensitivity, this is because these nanomaterials have large surface area to volume ratios, which can not only increase gas heat conduction, but can also reduce solid heat conduction, but their preparation processes are not compatible with CMOS processes and are not conducive to integrated mass production. Based on the comparison, we have found that different type MEMS Pirani sensors have different properties and are suitable for different application scenarios.

## 7. Applications

For some devices requiring a vacuum environment for operation, it is significant to monitor the vacuum degree after they are encapsulated, therefore, a vacuum detection sensor needs to be integrated into the device sealing can. Table 2 summarizes the pressure requirements of several different equipment or sensors when they are working. According to the pressure requirements of these equipment and sensors, different type MEMS Pirani sensors can be used to monitor their working pressure.

Through Table 1 and Table 2, it is possible to find out the performance differences of different type MEMS Pirani sensors and their possible application scenarios. It can also be found that for MEMS Pirani sensors with a lower detection limit ≤10^−3^ Pa, the preparation process is not compatible with CMOS processes, while for MEMS Pirani sensors that are compatible with CMOS processes, the detection range is not wide enough, which greatly limits the practical applications of the MEMS Pirani sensors. Based on this, it is important to integrate new materials or expand new structures to increase gas heat conduction and reduce solid heat conduction, broaden the detection limits and increase the operating range of the MEMS Pirani sensors on the premise that they are compatible with CMOS processes and can be produced in large scale at low cost.

MEMS Pirani sensors allow real-time monitoring of the reliability of vacuum encapsulated devices [114]. Figure 9a is a schematic diagram representation of the application scenario for MEMS Pirani sensors reported by Chen et al. [115]. In this device, two identical sensors are encapsulated in a single chip with a silicon cap wafer. A wall between the two sensors divides them into two separated microcavities. In addition, an air-path in the cap was constructed to open one of the microcavities to the ambient environment, generating two different pressures in one chip (one at 1 atm and the other one is unknown). Then, the sensor at 1 atm was used as a reference device to extract background noise, size variations and doping differences at the 1 atm ambient pressure. On the other hand, the other sealed device was used to measure the unknown cavity pressure. The difference between the output results of these two identical Pirani sensors was used to improve accuracy of the measurement results. Figure 9b demonstrates details of the device. Figure 9c shows normalized resistance change versus pressure at different time and different wafers as well as different lots, the results illustrate that the output consistency of the Pirani sensor was quite high under different conditions, indicating that the device has outstanding stability.

Figure 9d,e demonstrate the applications of the differential Pirani sensor. Figure 9d shows the results of 50 chips embedded in a wafer used for analysis of the pressure distributions across the wafer and for checking quality of the packaging. Measurements of wafer pressures illustrate that most of the final pressures in the cavity were in a range of 10–40 Torr (about 1333–5332 Pa), but the cavities in the center have final pressures of 760 Torr (about 10^5^ Pa), indicating that they were poorly sealed. However, after 96 h of wafer-level uHAST (un-biased High Accelerated Stress Test) tests, there were more cavities in the central part showing abnormal pressure values, indicating that the bonding degradation occurred in this area. Figure 9e shows the relationship between the five different bonding methods (#1, #2, #3, #4 and #5) and the final pressure in the wafer cavity, the results demonstrate that using bonding methods #1, #2 and #3 can achieve normal pressure of the cavity.

## 8. Conclusions and Outlook

In this review, different types of MEMS Pirani sensors are surveyed in detail, and the configuration, material, mechanism, and performance of different type MEMS Pirani sensors are systematically summarized and compared. Based on the Pirani effect, this review introduces the measurement principle and development status of various micro-Pirani sensors.

Due to their large sizes, traditional Pirani sensors are not suitable for rapidly developing microelectronics age. Therefore, in order to meet more development needs, a variety of miniaturized Pirani sensors have been created. According to the measurement principle, Pirani sensors can be classified into four types, including the thermistor-based Pirani sensors, the thermocouple-based Pirani sensors, the diode-based Pirani sensors and the SAW Pirani sensors. Finally, the development of the existing Pirani sensors adopting functional materials is introduced.

In thermistor-based Pirani sensors, the sensing element is the thermistors, thus temperature variations will cause changes of their resistance. Based on this principle, such devices are able to achieve a measurement range of 10^−1^–10^5^ Pa. Besides, such devices can be classified into two types according to the heat-transfer directions: the vertical and lateral configurations. The thermocouple type of sensors is based on the Seebeck effect, and can reach a measurement range of 5 × 10^−3^–10^5^ Pa. The diode ones are able to take function because the pn-junction output voltage varies corresponding to the vacuum degree, the measurement range of the devices can reach 2 × 10^−3^–10^5^ Pa. The SAW Pirani sensors are able to sense the vacuum information based on the SAW frequency change caused by temperature variation. For this type of sensors, a wide measurement range of 10^−3^–10^5^ Pa can be achieved.

Pirani sensors taking use of functional materials, such as porous AAO, CNTs and graphene, are not only simple structured, but also have excellent advantages in measurement range and response speed. The measurement range of these devices can even be extended to 10^−4^–10^5^ Pa. Inspired by this, the authors believe that there is a promising direction to further improve performance of the Pirani sensors, that is to develop novel functional materials which can be integrated into the Pirani sensors. These functional materials have large surface area to volume ratios, which can not only increase gas heat conduction, but also can reduce solid heat conduction. Thus, even if pressure is very low, changes in pressure can still be reflected by gas heat conduction. Of course, there are other ways that can help with the development of the Pirani sensors, including the demand of designing new structures and more suitable manufacturing processes that are compatible with CMOS processes. With all these efforts, MEMS Pirani sensors that have high performance and can meet different demands are expected.

## Figures and Tables

**Figure 1 micromachines-13-00945-f001:**
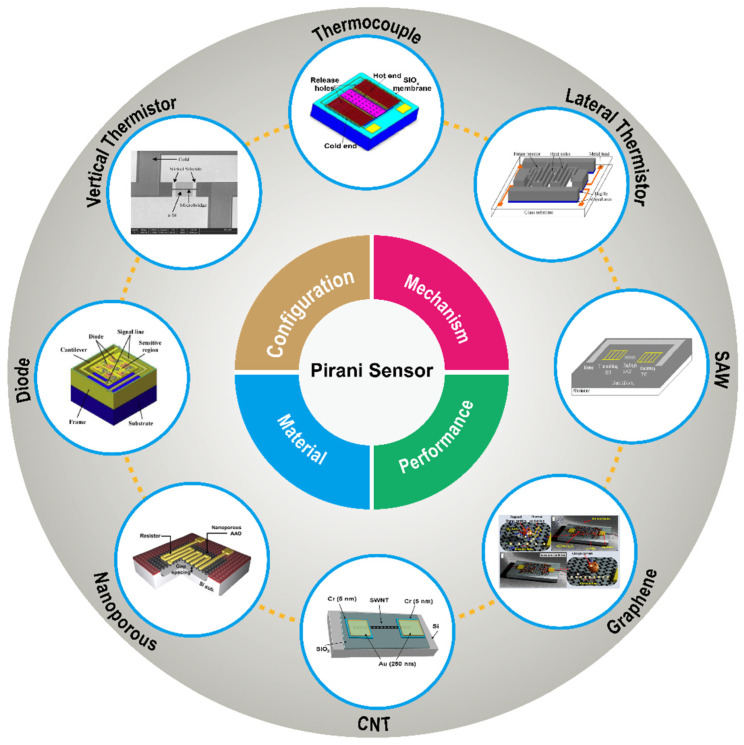
A summary about different types Pirani sensors.

**Figure 2 micromachines-13-00945-f002:**
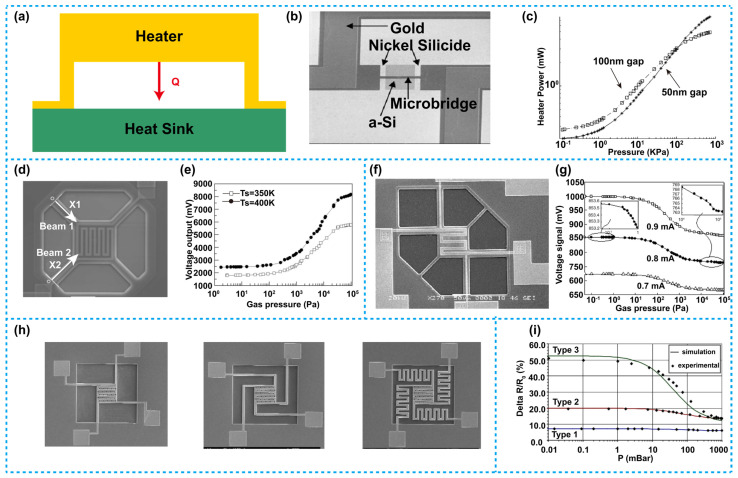
Thermistor-based Pirani sensors with a vertical heat transfer configuration. (**a**) Schematic diagram of such a Pirani sensor. (**b**) A sensor using a NiCr microbridge, such a device has an extremely narrow gap between its heater and the heat sink [66]. (**c**) Heater power variation versus pressure for the sensor in (**b**), the device with a smaller gap between its heater and the heat sink has a larger sensitive range and higher sensitivity. (**d**) A Pirani sensor based on a dielectric membrane structure with four supporting beams [73]. (**e**) The output voltage versus pressure of the sensor in (**d**), its sensitive range was 10 to 10^5^ Pa. (**f**) A Pirani sensor based on a dielectric membrane structure with six supporting beams [74]. (**g**) The output voltage versus pressure of the sensor in (**f**), its sensitive range was 10^−1^ to 10^5^ Pa. (**h**) A dielectric membrane-based Pirani sensor with three different heater areas [72]. (**i**) The resistance variation versus pressure of the sensors in (**h**), the device with a larger heater area has higher sensitivity and a wider sensitive range than that with a smaller heater area.

**Figure 3 micromachines-13-00945-f003:**
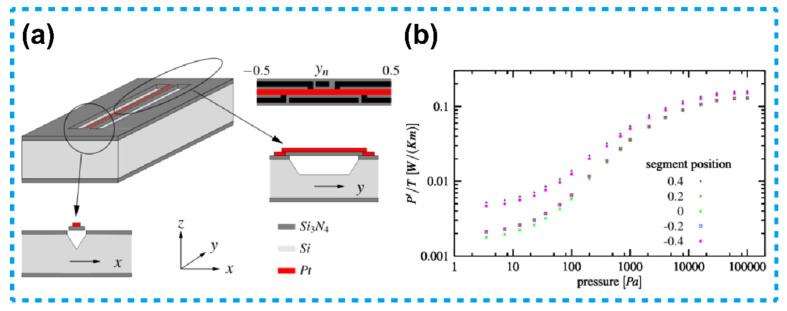
A vertical heat transfer configuration thermistor-based Pirani sensor with a V-shaped groove. (**a**) Schematic diagram of the sensor. (**b**) The power per unit temperature versus pressure, the sensitive range of the device was 3 to 10^5^ Pa [75].

**Figure 4 micromachines-13-00945-f004:**
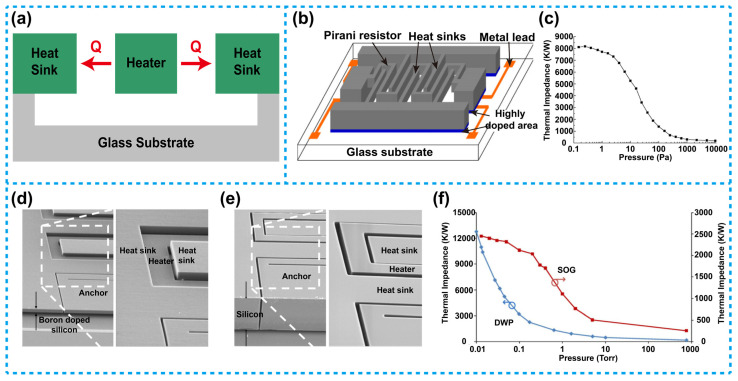
Thermistor-based Pirani sensors using the lateral heat transfer configuration: (**a**) Typical schematics of such a sensor. (**b**) Structure diagram of such a sensor [78]. (**c**) Thermal impedance versus pressure of the sensor mentioned in (**b**), the sensitive range of the device was 1 to 1000 Pa. (**d**) SEM image of the DWP sensor. (**e**) SEM image of the SOG device [79]. (**f**) Thermal impedance versus pressure of the DWP and SOG sensors mentioned in (**d**,**e**), showing the different sensitive ranges of the two devices.

**Figure 5 micromachines-13-00945-f005:**
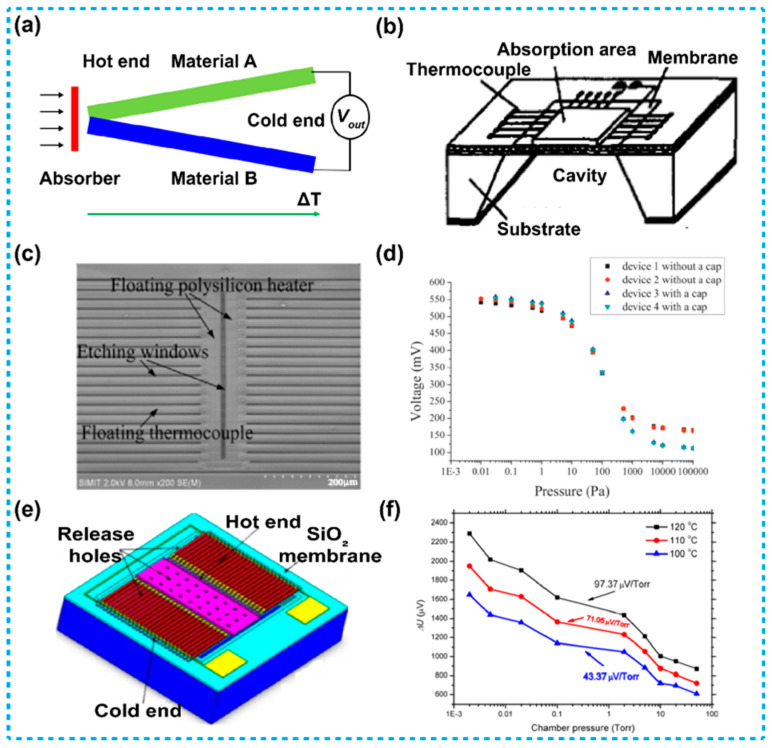
Thermocouple-based Pirani sensors. (**a**) A schematic diagram illustrating working principle of the Seebeck effect. (**b**) A schematic diagram of a conventional thermopile device. (**c**) Diagram of a thermocouple-based Pirani sensor with a heater located beside the hot ends [89]. (**d**) The output voltage versus gas pressure of the Pirani sensor mentioned in (**c**), the sensitive range of the device was 5 × 10^−3^ to 10^5^ Pa. (**e**) Diagram of a thermopile used for constructing a thermocouple-based Pirani sensor [90]. (**f**) The output voltage versus gas pressure at different heating power of the Pirani sensor mentioned in (**e**), the sensitive range of the device was 10^−1^ to 10^4^ Pa.

**Figure 6 micromachines-13-00945-f006:**
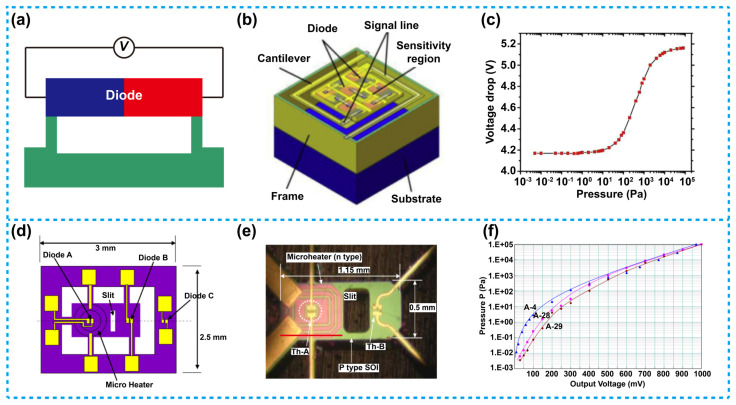
Diode-based Pirani sensors. (**a**) A schematic view of the diode-based Pirani sensors. (**b**) Schematic of a diode-based Pirani sensor based on a series of diodes [91]. (**c**) Voltage as a function of vacuum pressure; the sensitive range was 10^−1^ to 10^4^ Pa. (**d**) Schematic and (**e**) Micrographs of a diode-based Pirani sensor with diode-thermistor group combined with a heater [92]. (**f**) Relationship between pressure and output voltage difference for the sensor in (**e**), the sensitive range of the device was 2 × 10^−3^ to 10^5^ Pa.

**Figure 7 micromachines-13-00945-f007:**
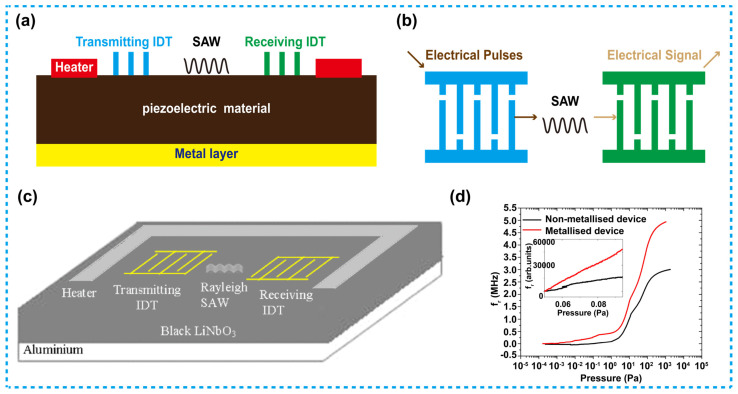
SAW-based Pirani sensors. (**a**) Schematic of a SAW Pirani sensor. (**b**) Structure of the interdigital transducers. (**c**) Schematic diagram of a metallized SAW Pirani sensor [98]. (**d**) Frequency response as a function of pressure observed for the nonmetallized and metallized devices, the sensitivity of the metallized device is higher than that of the nonmetallized device.

**Figure 8 micromachines-13-00945-f008:**
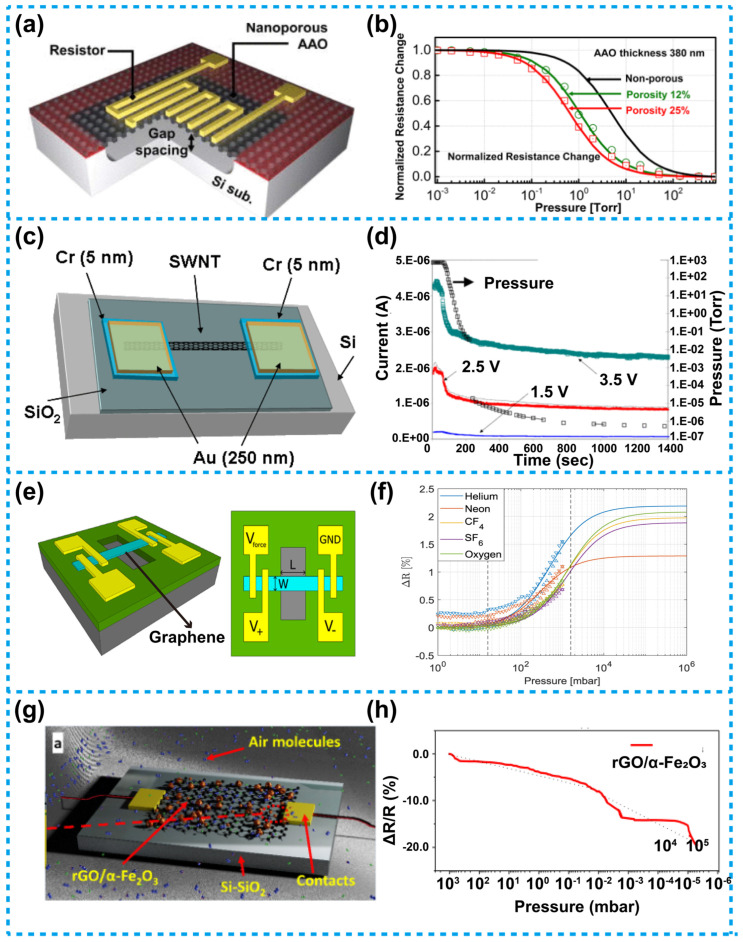
A Pirani sensor based on functional materials. (**a**) Schematic of the nanoporous AAO-based Pirani sensor [99]. (**b**) Normalized resistance of the sensor versus pressure, with a 380 nm thick AAO membrane, the curve shifts to the left, indicating a better ability to detect low pressure. (**c**) Schematic of a Pirani sensor based on SWNT [100]. (**d**) Dynamic pressure response of device in (**c**), the sensitive range reaches 10^−6^ to 760 Torr (about 10^−4^ to 10^5^ Pa). (**e**) Schematic of a sheet graphene-based Pirani sensor [103]. (**f**) Schematic diagram of resistance variation rate of the suspended graphene versus pressure in different gas circumstances. (**g**) Schematic and principle of Pirani sensors based on rGO-αFe_2_O_3_ [104]. (**h**) Resistance variation rate of the device versus pressure, the sensitive range was about 4 × 10^−6^ to 10^3^ mBar (4 × 10^−4^ to 10^5^ Pa, 1 mbar = 100 Pa).

**Figure 9 micromachines-13-00945-f009:**
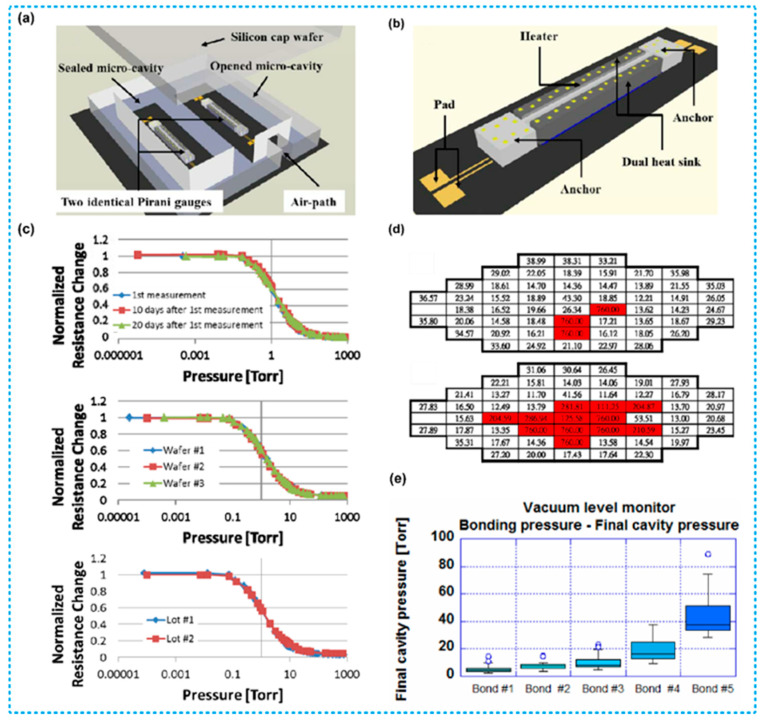
MEMS Pirani sensors for monitoring wafer-level packaging [115]. (**a**) Schematic diagram of a differential Pirani sensor. (**b**) The magnified view of a differential Pirani sensor. (**c**) The normalized resistance changes versus pressure. (**d**) Pressures of a vacuum-sealed wafer before and after 96 h of wafer-level uHAST tests, and the red area mean poorly sealed. (**e**) The relationships between five bonding recipes with their final cavity pressures.

**Table 1 micromachines-13-00945-t001:** The properties of different type MEMS Pirani sensors.

Type		Lower Detection Limit	Upper Detection Limit	Min Active Area	Min Power Consumption	Max Average Sensitivity	Common Active Materials	CMOS Compatible
Thermistor	Vertical	10^−1^ Pa	7 × 10^5^ Pa	1.35 µm^2^	160 µW	3.11 mV/Pa	Metal/Silicon	Yes
	Lateral	1.333 Pa	10^5^ Pa	0.13 mm^2^	7 mW	12.85 mV/Pa	Silicon	Yes
Thermocouple		5 × 10^−3^ Pa	10^5^ Pa	0.1 mm^2^	1.1 mW	104 mV/dec	Metal/Poly-Si	Yes
Diode		2 × 10^−3^ Pa	10^5^ Pa	1225 µm^2^	50 µW	90 µV/Pa	Silicon	Yes
SAW		10^−3^ Pa	10^5^ Pa	2.4 mm^2^	125 mW	1.38 MHz/Pa	piezoelectric materials	No
Functional material	Nanoporous	10^−2^ Pa	10^5^ Pa	0.04 mm^2^	N/A	N/A	AAO	No
	CNT	10^−4^ Pa	10^5^ Pa	N/A	15 nW	8 KΩ/Pa	CNT	No
	Graphene	4 × 10^−4^ Pa	10^5^ Pa	10 µm^2^	8.5 mW	2.5%/dec	Graphene	No

**Table 2 micromachines-13-00945-t002:** Pressure requirements for different equipment or sensors.

Equipment/Sensors	Working Pressure
Reactive Ion Etching	≤10^−3^ Pa
Microbolometer	≤10^−2^ Pa
Gyroscope	10^−2^–10 Pa
Resonator	10^−2^–10 Pa
Radio Frequency Switch	10^−2^–10 Pa
Physical Vapor Deposition	10^−1^–10 Pa
Uncooled Focal Plane Array	≤5 Pa
Chemical Vapor Deposition	10^3^–4 × 10^4^ Pa
Accelerometer	3 × 10^4^–7 × 10^4^ Pa

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
