# Peer review of "Overview of the MEMS Pirani Sensors"

_micromachines, 2022, doi:10.3390/mi13060945_

Round 1
Reviewer 1 Report
The authors have addressed the reviewer comments from the previous round. I recommended this manuscript for publication.
Author Response
Dear reviewer:
We sincerely thank you for the insightful suggestions concerning our manuscript. These suggestions are very valuable for further improving the quality of this paper. In the revised manuscript, we have made changes in accordance with your suggestions. In the marked version of the revised manuscript, changes to the manuscript are indicated in a “Track Changes” mode. We hope that the quality of our manuscript has met the publication standard after this revision.
[Actions]
Based on the reviewers' suggestions, we have made the following changes to the English language and style of the manuscript.
In the revised manuscript, we have changed the phrase “to improve” to “improving” in Line 38 of Page 1 in the Introduction.
In the revised manuscript, we also have changed the phrase “decrease of” to “a decrease in” in Line 41 of Page 1 in the Introduction.
In the revised manuscript, we also have removed the word “is” in Line 138 of Page 4 in Section 2.1.
In the revised manuscript, we also have removed the word “an” in Line 139 of Page 4 in Section 2.1.
In the revised manuscript, we also have changed the word “easy” to “ease” in Line 217 of Page 6 in Section 2.2.
In the revised manuscript, we also have removed the word “being” in Line 250 of Page 7 in Section 3.
In the revised manuscript, we also have changed the word “require” to “requiring” in Line 289 of Page 9 in Section 4.
In the revised manuscript, we also have changed the word “of” to “in” in Line 302 of Page 9 in Section 4.
In the revised manuscript, we also have changed the word “with” to “to” in Line 322 of Page 10 in Section 4.
In the revised manuscript, we also have changed the word “distributing” to “distributed” in Line 408 of Page 13 in Section 6.
In the revised manuscript, we also have changed the word “were” to “was” in Line 431 of Page 13 in Section 6.
In the revised manuscript, we also have changed the word “have” to “has” in Line 469 of Page 14 in Section 6.
In the revised manuscript, we also have changed the phrase “this type of devices have” to “this type of device has” in Line 473 of Page 14 in Section 6.
In the revised manuscript, we also have changed the phrase “thus” to “and thus” in Line 475 of Page 14 in Section 6.
Reviewer 2 Report
Dear authors,
I have reviewed comments and changes made in the manuscript.
Author Response

(The authors gave the same response as above.)

Reviewer 3 Report
Dear Authors,
The manuscript titled "Overview of the MEMS Pirani sensors" has been submitted. Minor spell check may be required before being accepted.
Best
Author Response
Dear reviewer:
We sincerely thank you for the insightful suggestions concerning our manuscript. These suggestions are very valuable for further improving the quality of this paper. In the revised manuscript, we have made changes in accordance with your suggestions. In the marked version of the revised manuscript, changes to the manuscript are indicated in a “Track Changes” mode. We hope that the quality of our manuscript has met the publication standard after this revision.
[Actions]
Based on the reviewers' suggestions, we have made the following changes to the English language and style of the manuscript.
In the revised manuscript, we have changed the phrase “to improve” to “improving” in Line 38 of Page 1 in the Introduction.
In the revised manuscript, we also have changed the phrase “decrease of” to “a decrease in” in Line 41 of Page 1 in the Introduction.
In the revised manuscript, we also have removed the word “is” in Line 138 of Page 4 in Section 2.1.
In the revised manuscript, we also have removed the word “an” in Line 139 of Page 4 in Section 2.1.
In the revised manuscript, we also have changed the word “easy” to “ease” in Line 217 of Page 6 in Section 2.2.
In the revised manuscript, we also have removed the word “being” in Line 250 of Page 7 in Section 3.
In the revised manuscript, we also have changed the word “require” to “requiring” in Line 289 of Page 9 in Section 4.
In the revised manuscript, we also have changed the word “of” to “in” in Line 302 of Page 9 in Section 4.
In the revised manuscript, we also have changed the word “with” to “to” in Line 322 of Page 10 in Section 4.
In the revised manuscript, we also have changed the word “distributing” to “distributed” in Line 408 of Page 13 in Section 6.
In the revised manuscript, we also have changed the word “were” to “was” in Line 431 of Page 13 in Section 6.
In the revised manuscript, we also have changed the word “have” to “has” in Line 469 of Page 14 in Section 6.
In the revised manuscript, we also have changed the phrase “this type of devices have” to “this type of device has” in Line 473 of Page 14 in Section 6.
In the revised manuscript, we also have changed the phrase “thus” to “and thus” in Line 475 of Page 14 in Section 6.

This manuscript is a resubmission of an earlier submission. The following is a list of the peer review reports and author responses from that submission.
Round 1
Reviewer 1 Report
The article attempts to review the recent developments in MEMS pirani sensors considering various aspects. However, there are some concerns over the term “recent advancements” as I can see authors have cited a total of 113 references, out of which only 54 references are related to MEMS Pirani sensors. Further, out of 54, only 29 references are recent (2012-2022). Only 04 references are from year 2021, 03 refences are from 2020, 03 references are from 2019 and so on.
As per the trends, I can see that the article is not much researchable and thus, this review paper is not feasible now. Many review articles are available for Pirani sensors and I don’t find any newness in this paper.
Some of the observations from the theory reported, I can see that
Figure. 2 is taken from papers belonging to [2006-2010]
Figure. 3 is taken from papers belonging to [2009-2010]
Figure. 4 is taken from paper belonging to [2009, 2016]
Figure. 5 is taken from paper belonging to [2007, 2019]
Figure. 6 is taken from paper belonging to [2011]
Figure. 8 is taken from paper belonging to [2015]
Beside this, from the conclusion, it can be seen that the Pirani sensors are good but there are no future directives for research in this field. It is kind of report rather than a systematic review article.
I do not recommend this article for consideration as it does not add significant knowledge to the field and topic is very old as one of the articles cited from year 1931.
Reviewer 2 Report
I recommend this manuscript for publication after the abstract is revised. The current abstract does not sound grammatically correct and lacks quality of writing. There are sentences which are long, complex and have repetitions. They should be separated for better readability and making sure the abstract as a whole provides a good summary of the article. e.g. line 10-13.
Reviewer 3 Report
Dear authors,
I've attached a comment in a pdf file.
